# Biomechanical Modelling for Tooth Survival Studies: Mechanical Properties, Loads and Boundary Conditions—A Narrative Review

**DOI:** 10.3390/ma15217852

**Published:** 2022-11-07

**Authors:** Saúl Dorado, Ana Arias, Jesus R. Jimenez-Octavio

**Affiliations:** 1Department of Mechanical Engineering, Escuela Técnica Superior de Ingeniería ICAI, Universidad Pontificia Comillas, 28015 Madrid, Spain; 2Department of Conservative and Prosthetic Dentistry, School of Dentistry, Complutense University, 28040 Madrid, Spain; 3Instituto de Investigación Tecnológica, Escuela Técnica Superior de Ingeniería ICAI, Universidad Pontificia Comillas, 28015 Madrid, Spain

**Keywords:** biting, bruxism, clenching, characterization, experimental, FEA, modelling

## Abstract

Recent biomechanical studies have focused on studying the response of teeth before and after different treatments under functional and parafunctional loads. These studies often involve experimental and/or finite element analysis (FEA). Current loading and boundary conditions may not entirely represent the real condition of the tooth in clinical situations. The importance of homogenizing both sample characterization and boundary conditions definition for future dental biomechanical studies is highlighted. The mechanical properties of dental structural tissues are presented, along with the effect of functional and parafunctional loads and other environmental and biological parameters that may influence tooth survival. A range of values for Young’s modulus, Poisson ratio, compressive strength, threshold stress intensity factor and fracture toughness are provided for enamel and dentin; as well as Young’s modulus and Poisson ratio for the PDL, trabecular and cortical bone. Angles, loading magnitude and frequency are provided for functional and parafunctional loads. The environmental and physiological conditions (age, gender, tooth, humidity, etc.), that may influence tooth survival are also discussed. Oversimplifications of biomechanical models could end up in results that divert from the natural behavior of teeth. Experimental validation models with close-to-reality boundary conditions should be developed to compare the validity of simplified models.

## 1. Introduction

A proper characterization and evaluation of the human teeth at a structural level has been of great interest for the last 30 years. In addition to a great number of studies [1,2,3,4,5,6,7,8], several relevant reviews focused in the structural tissues of the tooth from different points of view [9,10,11,12,13]. As a result, researchers have provided a complete vision of the structure of these tissues on healthy teeth. Nonetheless, there are several properties and responses of these tissues that are yet to be fully comprehended to understand the complex biomechanical phenomena. Being able to provide an adequate representation of the structural response and mechanical properties of teeth is of the upmost relevance for future studies to provide the possibility of studying teeth behavior not only through experimental studies but also through finite element analysis using real human models.

So far, different approaches have been used to retrieve the necessary information. On the one hand, experimental studies have provided data regarding the mechanical response of the different tissues through compressive tests [4,14,15], fatigue tests [6,16,17,18,19] or bending tests [8,20,21], to name a few; as well as, describing the effects of different conditions (treatments, restorations, habits, etc.) on the resistance of the tooth as a whole under functional and parafunctional loads [22,23,24,25,26,27,28,29]. Functional loads can be defined as the natural and most common loads and motions performed by the human body, whereas parafunctional loads are associated to certain disorders or malfunctions of the body [30].

On the other hand, finite element studies have mostly aimed for simulating this latter experimental approach and studying the cases in a more precise manner, being able to analyze stress distributions, fatigue response or crack propagation, among other mechanical phenomena. Nonetheless, both approaches, through these last 30 years, have studied situations in which the natural biomechanical behavior of the tooth is not completely reflected. The difficulties of studying human samples or oversimplification of the mechanical and finite element studies, among several other factors, make the reproducibility of these situations difficult.

While the present literature describes the biomechanics of teeth from a global point of view, it is yet to be proven that teeth show the same mechanical response in simplified models and in the mouth. Therefore, the aim of this review was to identify the most relevant factors related to tooth survival that should be implemented in future dental biomechanical modelling studies in terms of mechanical properties of the complex tooth/periodontal ligament/bone, functional and parafunctional loads, and boundary conditions.

## 2. Mechanical Properties of Structural Tissues

A tooth is composed of several layers of hard tissue surrounding the dental pulp, which contains blood vessels, soft tissues, and nerves [31]. Being that the dental pulp is removed in many endodontic treatments, its structural properties are not relevant for the tooth’s structural integrity in endodontically treated teeth as it is neither a load bearer nor as load transmitter. Nonetheless, other than enamel and dentin, the periodontal ligament (PDL) is structurally relevant despite being a soft tissue external to the tooth, as its main functionality is to transmit the loads applied to the tooth to the jaw or maxilla while holding the tooth in place. Additionally, the bone (both trabecular and cortical) surrounding the tooth (whether it is in the mandible or the maxilla) is the main support of the tooth and the receiver of the loads transmitted by the PDL from the teeth.

Therefore, we can define the following structurally relevant tissues of the tooth: enamel and dentin internally, and PDL and trabecular and cortical bone externally. Enamel and dentin have been identified as the loadbearing tissues of the tooth itself [1,10,11], and it is generally accepted that the periodontal ligament transmits occlusal loads onto the alveolar bone [32,33].

Different relevant reviews have already summarized the properties and structures of these tissues [9,10,11,13,34] showing heterogeneous results due to the broad number of factors that can affect the state of a tooth, (among them the age [2,35] of the donors, their habits and health conditions) and the methodological differences from the different studies, mainly the precedence and conservation of the samples, the methods used for load application and the numerical loads applied.

Additionally, enamel and dentin fatigue properties and responses have also been thoroughly studied and summarized in former reviews [10,12,36,37,38,39]. Yahyazadehfar et al. (2014) presented a technical review including all kinds of fatigue and fracture testing performed until 2014, along with the obtained fatigue and fracture properties of both dentin and enamel [12]. Arola (2017) reviewed the different methodologies to perform fatigue tests on dental materials and restorations, reaching the conclusion that fatigue is one of the main contributors on the failure of dental restorations [37].

## 3. Enamel

Enamel is the first occlusal layer of dental hard tissue. It is composed of enamel rods perpendicular to the dentin–enamel junction [3]. Enamel is the first tissue to suffer load from contact with antagonist teeth, although the resistance of enamel depends greatly on the supporting dentin. For this reason, the mechanical properties of enamel have not been as thoroughly studied as those of dentin. Nonetheless, several studies have reported different results related to enamel properties.

Table 1 shows a description of the data provided for sample characterization, methodology used and results obtained by studies reporting on enamel mechanical properties. A consequence from the data provided was also included for each specific study. Unless otherwise specified the studies used were caries and damage-free samples. When available, sample size was also provided.

Shimomura et al. (2019) investigated the Poisson ratio of enamel, reporting a value of 0.303 ± 0.098 [49], which is in line with other studies in the literature [51,53].

Some studies have reported values for the threshold stress intensity factor (∆Kth), ranging between 0.35 and 0.4 MPa m^0.5^ [16,39]. Interestingly, Bajaj et al. (2008) analyzed the fatigue crack growths mechanisms of enamel and hydroxyapatite using Paris Law [16]; although later, the same author, Bajaj and Arola (2009) recommended not to apply Paris law on enamel as it is not applicable for characterizing non-linear cyclic extension in the short crack regime of the forward direction or for the reverse direction [54].

Gao et al. (2016) evaluated the fatigue response of human enamel by cyclic contact and found out that enamel has no apparent endurance limit in functional loads. Additionally, the contact fatigue response in their study exhibited a reduction in number of cycles to failure when increasing the magnitude of the cyclic load [55]. Yahyazadehfar et al. (2013) observed that the apparent endurance limit of enamel was approximately 12 MPa [56]. Yahyazadehfar et al. (2014) [12] also reviewed the fatigue and fracture properties of enamel, reporting similar values to those of Table 1 for fracture toughness (Kc) and similar values stress intensity threshold (∆Kth) to those obtained by Bajaj et al. (2008) [16] and Arola et al. (2010) [39].

Most results shown in Table 1 correspond to studies that have used indentation or nano-indentation to obtain the mechanical properties of enamel with in vitro samples. This is mainly due to the difficulty to obtain valid probes from tooth samples, as their size hinder the process of probe machining and the anisotropic distribution of the enamel rods make the results differ from one probe to another. The main limitations of indentation and nano-indentation studies performed with in vitro samples are related to the sample itself and its conservation since the non-ideal conditions between extraction and testing time could dry the samples, and hence, reduce their mechanical properties (see Boundary conditions for lifespan analysis). Additionally, the inclusion of samples from patients with different conditions increases intrasubject variability generating broad differences in the results.

## 4. Dentin

Below the enamel layer, there is a dentin layer. The dentin surrounds the dental pulp, and its structure extends perpendicular from the dental pulp towards the dentin enamel junction or the exterior cementum.

As mentioned before, the mechanical properties of dentin have been deeply studied compared to any other dental tissue. Nonetheless, there is still a significant heterogeneity in reported results on its properties.

Table 2 summarizes the data provided for sample characterization, methodology used, results obtained and consequences of the studies reporting on dentin mechanical properties. 

Different authors have reported a range of values for the Poisson ratio between 0.3 and 0.31 [15,52,56]. Moreover, for dentin, most results regarding the mechanical properties were obtained through indentation or nanoindentation (similar to enamel), thus sharing the same limitations. As for the fatigue properties, most studies based their methodologies on creating fracture test samples from dentin obtained from extracted teeth. As dentin is an anisotropic heterogeneous tissue, these studies also have several limitations already discussed [57,58].

Table 3 summarizes the results obtained by previous studies regarding dentin fatigue properties.

Most studies that have analyzed fracture toughness of dentin are based on linear-elastic fracture mechanics (LEFM). Whilst dentin has partly an elastic behavior, LEFM do not exactly describe the exact behavior of dental hard tissues. Yan et al. (2008) investigated the fracture toughness of dentin using elastic–plastic fracture mechanics (EPFM) and reported that plastic deformations required 60% of the energy spent on elastic deformations [20]. Yahyazadehfar et al. (2014) [12] also reviewed the fatigue and fracture properties of dentin, reporting similar values for both fracture toughness (K_c_) and stress intensity threshold (∆K_th_) than those included in Table 3.

At the same time, dentin response under cyclic loads has been thoroughly studied (see Table 3) and the endurance limit settled in 44–50 MPa range [19,61,62,63].

## 5. Periodontal Ligament

The PDL is external to the tooth. It is composed of aligned fibers attached on one side to the root cementum and on the other side to the alveolar bone of the jaw/maxilla. It is structurally relevant to the tooth as it absorbs part of the applied loads on teeth and provides mechanical stability. PDL fibers are not evenly aligned, their direction varies depending on the area of the root that they are attached to [33]. The elastic behavior of the PDL is influenced by the loading rate, tooth type, root level, and individual variation [7].

Table 4 summarizes some of the results obtained by previous studies regarding PDL mechanical properties, as well as data provided for sample characterization, methodology used and consequences. 

There is not a consensus on the mechanical properties of the PDL, as seen in Table 4. Former reviews have reported an elastic modulus and Poisson ratio ranging, respectively, from 0.01 to 1379 MPa and from 0.28 to 0.49 [32]. Although there is no consensus either in the values for dentin or enamel, the ranges are narrower (see Table 1 and Table 2) than those found in PDL. This is mainly due to the characteristics of the tissues. While enamel and dentin are hard tissues, which decay is slow, and have an elastic–plastic behavior, ligaments are soft tissues which behavior has been often described as viscoelastic [68,69,70], thus making it a difficult material to characterize due to its strain rate-dependent response. Moreover, the connective nature of the tissue makes it more difficult to preserve out of the mouth of patients. Specifically, PDL mechanical response is not yet clearly defined. While some studies describe its behavior as viscoelastic [68,71,72,73], others describe it as hyperelastic [74,75] or visco-hyperelastic [76,77]. As well as other ligaments in the human body [78], PDL response to cyclic loads is dependent on the loading frequency and the number of cycles. Wu et al. (2019) studied the effect of different frequencies on the viscoelastic response of PDL, indicating that PDL viscoelastic properties depend on the loading frequency with an exponential function [72].

## 6. Bone (Mandible and Maxilla)

Although bone from both mandible and maxilla are completely external to the tooth (there is no interconnection between tissues, such as PDL), these osseous tissues are relevant for the teeth survivability, as they will absorb part of the load exerted to the teeth and act as their support. Maxillary and mandibular tissues are anisotropic [79,80,81,82,83], this means that their mechanical properties will be different depending on the analyzed direction.

Additionally, cortical bone and trabecular bone have different mechanical properties and are characterized separately. Table 5 summarizes some of the results obtained by previous studies on the mechanical properties of mandible and maxilla, as well as data provided for sample characterization, methodology used and consequences.

Most studies in Table 5 described the anisotropic properties of both trabecular and cortical bone [79,80,86,87], finding significant differences among studies, specifically among results obtained with compression tests. This difference can be due to the different methodologies of sample preparation. There is little information on the compression strength of the mandibular bone. Kido et al. (2011) reported values of 237, 216 and 196 N for the ultimate compressive load of anterior, premolar and molar trabecular mandibular bone. Misch et al. (1999) reported a compressive strength of 5.38, 2.57 and 1.70 MPa for the ultimate compressive load of anterior, premolar and molar trabecular mandibular bone.

As shown in Table 5, to the current authors’ knowledge, there is scarce information on the mechanical properties of maxillary bone, especially on the anisotropy of trabecular bone.

Several studies have measured different mechanical properties in maxilla and mandible taking different measurement points throughout the bone [80,81,82]. Being these measurements punctual and being demonstrated that mechanical properties in these bones depend on the site of the sample, infinite measures would be necessary for a complete and perfectly accurate model of the bone. Despite this, there are other approaches that can be used to approximate the model to reality. Natali et al. (2010) suggested an interpolation method with a function dependent on the angle and longitudinal position of each point relative to the distal-mesial axis [88].

While it is generally accepted that both trabecular and cortical maxillary and mandibular bone is anisotropic, it has been a common practice in the former literature to consider bone as isotropic to simplify computational models because most studies disregard the important role of bone focusing only on the properties of teeth. On the contrary, dental cementum does not have a relevant structural role and for this reason is often not included in computational experiments and its mechanical properties are described by a scarce number of studies [1,89,90,91,92].

The lack of a consensus for the properties of structural tissues is probably due to the heterogeneity of conditions of both samples and experiments. It is difficult to find studies with a sufficient number of samples with the exact same conditions (age, gender, habits, etc.). Additionally, the results may vary depending on the conservation of the samples. Samples obtained from extracted teeth might have modified mechanical properties with time [93], which could be further affected with repetitive freeze-thaw cycles [69,94,95] and lack of moisture during experimentation [96].

## 7. Functional and Parafunctional Loads

Once the mechanical properties of the different tissues have been addressed, the second group of relevant parameters to consider in biomechanical studies is the consideration of the different loads that can occur within teeth. Jaw movements can take place in three different directions: Inferior-superior, left-right and antero-posterior. These movements, when teeth are in contact, cause different loads that can influence tooth lifespan. Specifically, the influence of loads in three different actions have been studied thoroughly in different studies: Biting (functional load), clenching, and grinding (last two are often referred to as parafunctional loads or bruxism [97,98]).

Finite element methodologies (FEM) are common tools for mechanical studies. In biomechanical studies, these methodologies are used to validate (material properties, experimental tests, lifespan, etc.) or analyze (effect of several parameters on the structure of a tissue, prosthesis, etc.) [99,100]. In dental biomechanics, FEM have become an important tool for testing and researching, due to the size of teeth and the reduced viability for obtaining a big number of samples for statistical validation, as teeth are human, non-regenerative tissues [22,75,84,101,102,103].

The effect of each kind of load on teeth has been thoroughly studied from different points of view: from experimental studies analyzing the maximum loads until fracture or the fatigue properties of teeth [24,26,104,105] to FEM studies evaluating the critical load points of the tooth [25,26,102,106,107]. Both experimental and simulation approaches should be considered when studying the influence of these loads on teeth structure, as the information extracted from both approaches gives necessary information to understand the models at all levels.

Then again, regarding teeth lifespan, FEM have also been used to analyze the response of the tooth against cyclic loads (fatigue), which are closest to the natural behavior of the human tooth. Some studies have evaluated the effect of each type of load using finite element analysis and cyclic/dynamic loading [22,102]. Methodologies in FEA studies sometimes involve studying fracture strength and teeth lifespan under different conditions, from healthy teeth to restored pulp-less teeth. Soares et al. (2015) reported that the loading type not only influenced the biomechanical behavior of the maxillary premolars but also had a greater influence than several pathological conditions [108]. Therefore, the type of loading will be a very relevant factor when analyzing teeth lifespan. For this reason, a detailed description of the kinematics and forces during functional (biting) and parafunctional loads (clenching and grinding) follows separately for the three entities.

## 8. Biting

Once teeth encounter their antagonists during mastication, they experiment bite loads. These loads do not only appear in mastication processes, as the occlusal loads are mainly bite forces. If these occlusal loads do not involve an extraordinary recruitment of muscle fibers [109,110], thus applying loads over the functional behavior of the jaw, we will define them as biting occlusal forces.

Several studies have attempted to define the human masticatory movement frequency and number of masticatory events. Po et al. (2011) reported that human masticatory movement frequency ranges from 0.94 Hz to 2.17 Hz [111]. Hasegawa et al. (2009) studied the effect on brain activity of chewing tasks, finding that the average gum chewing frequency was 1.26 Hz [112]. Bessadet et al. (2013) obtained a masticatory frequency range (Chewing Frequency) between 1.44 Hz and 1.63 Hz [113]. Most reported masticatory frequencies are inside the interval defined by Po et al. (2011) and many are close to the interval defined by Bessadet et al. (2013). The amplitude of this range is probably due to the influence of body condition and chewed materials on chewing frequency [114,115].

Jaw trajectory during mastication has been thoroughly studied in the former literature. Occlusal loads during mastication are mainly directed in the vertical direction, but due to its trajectory not being vertical and having an incidence angle before contact in every direction, occlusal loads have small lateral and antero-posterior components [116,117,118,119]. Wang (2019) stated the kinematics and trajectory planning of a chewing robot which can reproduce human chewing trajectories [120]. Ogawa (2001) measured the incidence angles of contact between teeth during a chewing cycle, obtaining a closing angle of 72.5° ± 9.4° for vertical chewing and 46.6° ± 7.4° [121] for lateral chewing. This result is in line with the range of 15°–30° of incidence angle reported by several studies [119,120,122,123,124]. Bishop (1990) reported that the trajectory of the jaw is independent of the chewing frequency and the hardness of the chewed bolus [125].

Several papers have studied the effect of masticatory/biting loads on teeth over time. The effect of this type of load is often studied with cyclic loads that aim to reproduce the low force exerted between tooth during a normal mastication event, which has been reported to range between 50 N and 150 N [126,127,128,129,130,131,132,133], and their periodicity.

Table 6 summarizes former studies that analyzed the influence of biting forces on teeth.

All studies analyzed in Table 6, except Ossareh et al. (2018) [23], who used a chewing simulator, tend to simplify, using vertical or very angulated loads, the biting trajectory of the mandible-jaw system. Additionally, various studies using cyclic loads used frequencies out of the functional range of human mastication [104,105,135,138,145,147].

## 9. Clenching

Like biting occlusal forces, clenching occurs once teeth encounter their antagonists during an occlusal movement. The difference between biting occlusal forces and clenching occlusal forces is the bigger magnitude of the applied load. This difference is due to the muscle fibers recruited, being higher for clenching than for biting [109,110]. Therefore, we should consider clenching loads as parafunctional loads significatively higher than biting loads. In former studies, the effect of the “Maximum voluntary bite force/load”, “Maximum occlusal force/load” or “Maximum bite force/load” (MBF) in teeth has been studied. All these forces respond to the same concept of clenching; therefore, they will be analyzed as clenching loads in the present manuscript.

The frequency of these loads has not been thoroughly studied. As clenching loads are not repeated in a continuous way and they rather have a pulsating behavior separated with time, no specific frequency of clenching cycles per minute can be established. A study by Cioffi et al. (2017) reported that clenching episodes have a duration between 0.7 s and 1 s [150]. In this study, healthy participants exhibited a frequency of clenching episodes (>30% of right masseter maximum voluntary contraction) 10 times smaller than participants with masticatory muscles myalgia.

Regarding the displacement that happens during clenching, the main component of this applied load is vertical, as it is applied once teeth are in contact with their antagonists. Once a clenching episode takes place, the maxillary teeth tend to move in the palatal direction due to the shape of the teeth until they reach the total occlusal contact [151]. Once they are positioned, there should not be relative displacements between teeth, as the applied loads are compressive loads, which, along with the concave shape of the occlusal face of teeth crowns, will avoid the relative movement of teeth. Kawaguchi et al. (2007) measured the components and direction of the clenching loads in endodontically treated second molars, obtaining a vector from the crown to the root with an angle of 10° from the sagittal plane and 3° from the horizontal plane [152].

Clenching loads can be considered in between a broad range of values. This range includes loads above the normal masticatory forces (see Biting) and below the MBF. Therefore, it is necessary to establish the upper limit of this range. MBF values are dependent on several physical and physiological conditions [153]. Therefore, reported values are not homogeneous. The reported values usually range between 400 N and 1000 N [127,132,154,155,156,157,158,159,160], but some studies indicate that there are subjects able to go beyond that range [130,153]. Raadsheer et al. (1999) measured the components of the resultant maximum voluntary bite force in both men and women using a force transducer capable of registering forces in all three dimensions, finding that occlusal loads are not reduced to a single plane, but have three directional components [122].

Table 7 summarizes former studies that analyzed the influence of clenching forces.

Again, only a few studies of the ones analyzed in Table 7 did not simplify, using vertical or very angulated loads, the occlusal contact between maxillary and mandibular teeth [25,165,176]. Additionally, most studies using cyclic clenching loads were not able to reproduce the pulsating behavior of clenching loads, either using an incorrect frequency or maintaining the load for an excessive amount of time, finding only a small number that were able to do so [26,146,161,164].

## 10. Grinding

Similar to the previous conditions, grinding loads are parafunctional loads that involve teeth encountering their antagonists in an occlusal movement. After this contact happens, a compressive occlusal load is applied, such as a clenching load. The main difference between grinding and clenching is the relative frictional displacement between mandible and maxilla present in grinding loads. Although this frictional displacement can take different patterns [179], lateral (left-right) grinding [180,181,182] exerts lateral loads and friction to the occlusal surface of the tooth that cannot be effectively dissipated [183], presenting the highest risk for teeth lifespan.

The existing literature, up to date, does not offer a clear consensus on the frequency of grinding events, as they do not happen in a continuous nor cyclic manner per se [97,98,180,181,182,184,185,186]. Therefore, grinding events could be temporally defined in terms of an average duration of the event. Nonetheless, several studies have estimated a duration superior to 2 seconds of sleep bruxism grinding events (included inside tonic events) with reported values between 1 and 8 seconds [29,97,181,184,185,186,187,188,189].

Maximum laterotrusion (lateral movement of the mandible) has been reported to range between 7 mm and 12 mm [190,191]. Teeth grinding does not necessarily involve maximum laterotrusion. Teeth grinding associated to bruxism is the focus of interest for tooth lifespan due to the relevant values of the applied forces, as they will take values comparable to MBF due to the clenching motion. During sleep bruxism grinding events, several authors have reported lateral displacement values of the jaw that range between 1 mm and 5 mm [181,184].

The load exerted between teeth during grinding episodes is not well described in the literature. Nonetheless, the muscle activity has proven to be greater in grinding events than in clenching events [192,193], probably due to the sum of clenching loads with lateral excursions, which necessarily recruits more muscle fibers. Giannakopoulos et al. (2018) recorded the muscle activity and exerted force between jaw and maxilla with an intraoral device, obtaining a mean single resultant force vector during grinding episodes that was almost parallel to the frontal plane, with an angle of 10 to 15 degrees from the midsagittal plane [193]. To the best of the authors’ knowledge, there are no studies that support specific values or ranges for force values generated between teeth during grinding. Due to this lack of knowledge, some studies have aimed for reproducing specific subject motions or muscle forces (see Table 8).

Table 8 summarizes former studies that analyzed the influence of grinding forces.

Studies in the current literature have not established a criterion of reproducibility of grinding loads. As mentioned before, several studies have tried to reproduce them using recordings of patients to mimic the trajectories and loads exerted by the patients themselves. These studies, along with a better understanding of the grinding motion, should serve as the parting point for a better definition of experimental and FEA studies involving grinding loads.

## 11. Boundary Conditions for Lifespan Analysis

Being the influence on teeth lifespan of the type of load already demonstrated by the former literature and analyzed, we should consider that it is not the only relevant condition to take into account when analyzing the survivability of teeth. There are several factors, apart from the type of load, that can affect tooth lifespan. These factors range from physical and physiological conditions (age, humidity, etc.) to the selected tooth and the number of cycles and frequency applied. These factors can be classified in three groups: tooth, patient and environment-dependent factors.

## 12. Tooth Dependent Factors

When performing a study on a human tooth using cyclic loads, it is necessary to consider both the amplitude and the loading frequency. Each type of load has a reported range in its possible values (see Biting, Clenching and Grinding). Thus, it is necessary to adapt each load, regardless of its type, to the specific conditions of the samples.

Due to the mechanical response of the jaw, which resembles a lever [195], bite force is not distributed equally among teeth [196,197], being greater the more posterior the tooth is [198]. This phenomenon explains the bigger sizes of posterior teeth and their surrounding PDLs in comparison with anterior teeth, as they withstand bigger loads [198]. Additionally, there are teeth that tend to fracture with a higher frequency than others, being the first mandibular molars the most likely to present a fracture [199].

Choosing the right loading frequency for cyclic loads is also important due to the frequency-dependent fatigue response of structural tissues. Nalla et al. (2003) reported that human dentin presents a substantially longer life at 20 Hz than at 2 Hz, similar to the response of bone [6], which is typical of materials susceptible to creep. PDL is also dependent on the loading frequency (see Periodontal Ligament). No studies, up to the authors’ knowledge, have specifically described a frequency-dependent behavior of enamel. Nonetheless, Madini et al. (2015) studied the mean cuspal deflection of intact, filled with composite after endodontic treatment and filled with composite after endodontic treatment subjected to cyclic loading at 2 Hz, finding out that maximum mean cuspal deflection was present in the samples subjected to cyclic loads [200]. Therefore, applying loading frequencies close to the functional mandibular frequencies should not alter the natural behavior of the structural tissues of teeth and should give a more accurate representation of the teeth biomechanical response.

## 13. Patient Dependent Factors

In order to quantify fatigue tests results, one should be able to establish a connection between the number of loading cycles performed until desired results and the real time those cycles would translate into. Former studies do not agree on a specific number for mastication cycles per year, describing a range from 2.5 × 10^5^ to 1 × 10^6^ cycles per year [10,146,201,202,203,204].

Other physiological factors should also be taken into account for a good interpretation of the obtained results. Being healthy teeth living tissues, they tend to adapt to their environment and align their internal structure to withstand the most frequent loads exerted to them [205,206]. For that reason, factors such as age, gender, humidity and loading angle (see Biting, Clenching and Grinding) need to be considered when trying to reproduce the natural behavior of dental tissues.

The age of the specimen conditions the structural state and its response. Apart from broadly known facts, as lesser regeneration and tissue decay with age, there are other factors related to age that affect teeth lifespan. Mastication mechanisms are subject of changes with age [207]. Shinogaya et al. (2001) reported a lower occlusal force and higher contact area between teeth in senior subjects compared to younger subjects [208]. Recent studies also present results in accordance [209,210,211,212]. Mechanical properties are also affected by the age of the specimen. Dentin rate of damage initiation and propagation are higher at higher ages [17,35], while its fracture toughness decreases with age [17,57,59,213,214].

Gender, as well, is a relevant characteristic towards the total occlusion force that can be exerted between teeth. The magnitude of the MBF is different between genders [211]. Braun et al. (1996) studied the changes in MBF during growth in both male and female subjects from 6 to 20 years old, finding out that during growth, MBF increases in both males and females in a similar manner, but after puberty, male maximum bite forces increase at a greater great than females [215]. Shinogaya et al. (2001) reported a higher MBF and occlusal contact area in males than in females [208]. Recent studies also present results indicating that MBF is greater in adult and elder males than females [209,210,212,216].

Ethnicity is also a relevant factor towards the magnitude of MBF. Several studies indicate that MBF has significant variations in value between different ethnicities. Shinogaya et al. (2001) reported higher MBF values in Japanese females than in Danish females [208]. Shinkai et al. (2001) reported higher MBF values in European-American subjects than in Mexican-American subjects [217]. Borie et al. (2014) reported higher MBF values in Mapuche indigenous Chileans than in non-indigenous Chileans [218]. Psychological factors, such as stress, anxiety and depression, also impact the frequency in which temporomandibular disorders appear in patients [219].

The presence of artificial dentures (partial or complete) also conditions remaining teeth lifespan. Several studies indicate that MBF decrease for teeth loss is not completely reverted by dentures, being MBF bigger the smaller the denture [210,211]. Endodontically treated patients present higher MBF values due to sensitivity loss [220]. Additionally, missing mesial or distal adjacent teeth affects the survivability of the tooth [221].

## 14. Environment Dependent Factors

The human buccal cavity is lubricated and kept in wet environments thanks to the oral mucosa and saliva secreted by the oral glands and the mucosa membrane. Keeping this moisture is relevant for teeth lifespan, as the occlusal surfaces will keep lubrication during contact. Salivary lubrication and efficiency can affect tooth wear [97,133,222]. Using artificial saliva in biomechanical tests using teeth helps reproducing the natural conditions of the tissues [223]. Additionally, some studies have proven that the mechanical properties of hard tissues diminish after the tooth has its pulp removed due to it becoming dry. Huang et al. (1992) reported that both Young’s modulus and proportional limit in compression are higher in dentin from normal vital teeth than those from wet treated pulp-less teeth [96].

Other aspects such as personal hygiene [19] or smoking habits [224] can condition the state of the teeth and even their mechanical properties. Ibrahim and Hassan (2021) studied the effect of smoking habits on enamel, finding out that there is a decrease in microhardness and calcium in smoking subjects’ teeth [224].

## 15. Conclusions

A summary on the most common experiment conditions in biomechanical studies on teeth was presented, focusing on the most relevant factors for building a realistic model to study the biomechanical response and survivability of teeth (mechanical properties, loads and physiological and environmental considerations) that may influence the results of both experimental and finite element studies.

A good definition of the mechanical properties of structural tissues is most relevant for both understanding the behavior of tissues and its reproduction in FEA studies. Most mechanical properties fall into a range of values that is coherent with the heterogeneity of methodologies and samples. The following values have been reported for the different internal structural tooth tissues: enamel (Young´s modulus = 60–150 GPa; Poisson ratio = 0.3; compressive strength = 370–384.5 MPa; threshold stress intensity factor = 0.35–0.4 MPa m^0.5^; fracture toughness = 0.37–2.05 MPa m^0.5^) and dentin (Young´s modulus = 11.5–23.3 GPa, Poisson ratio =0.31; compressive strength = 248–300 MPa; threshold stress intensity factor = 0.5–1.23 MPa m^0.5^; fracture toughness = 1.13–2.5 MPa m^0.5^), The unclear mechanical response of the PDL is associated to higher range values (Young´s modulus = 0.01–1379 MPa; Poisson ratio = 0.28–0.49). Finally, the mechanical properties of maxillary and mandibular bone are dependent on the direction, thus, they present an anisotropic behavior in their Poisson ratio and Young Modulus in both cortical (Young´s modulus = 5–26.28 GPa; Poisson ratio = 0.22–0.45) and trabecular (Young´s modulus = 0.11–67.48; Poisson ratio = 0.3) tissues.

Both experimental and FEA studies that aim to study teeth biomechanics and how they change under different conditions and treatments (when necessary) have opted for different study conditions regarding the applied loads. Some have applied functional loads, while others have tried to reproduce the response under parafunctional loads to test the fracture strength or the fatigue response of the samples. Specifically, FEA studies have tried to reproduce the experimental response of the tooth, healthy or under different treatments, to study the effect of functional and parafunctional loads on the different areas of the tissues. The mechanical properties for the structural tissues used in these studies fall into the formerly mentioned ranges obtained in experimental studies.

Most experimental studies applying biting or clenching loads chose cyclic loads ranging between 50 and 260 N at frequencies that range from 1.2 to 6 Hz for biting loads and loads between 340 and 2800 N at frequencies that range from 0.072 to 20 Hz for clenching loads. Most applied loads for both clenching and biting experimental studies have applied vertical loads with only a few studies applying loads with 30° or 45° angles. FEA biting forces studies used loads between 10 and 200 N, mostly being applied vertically and some with angles between 35° and 45°. FEA clenching forces studies used loads between 340 and 1000 N, mostly being applied vertically and some with angles between 30° and 60°. FEA studies mainly focused on analyzing stress distributions, with some of them also analyzing fatigue response.

As for grinding loads, there are too few experimental studies to establish a range, therefore some studies have used recorded reaction forces during grinding events while others have assumed different points of load and frequencies. FEA grinding forces studies have mainly used jaw models to reproduce the lateral excursion or previously recorded loads, with very few information of loading conditions applied directly on teeth neither like loading nor like displacement.

Finally, while both mechanical properties and loading conditions are the main actors in biomechanical models, there are several environmental and physiological conditions that influence the state of the tooth, its survivability, and its response to the loads, such as age, gender, selected tooth, humidity, dental condition, or habits, among other parameters.

Attending to this information, a precise sample definition and characterization is a critical factor for defining an accurate biomechanical model, as applied loads are dependent on tooth, patient and environment dependent factors and FEA studies require an accurate definition of the material properties of the dental tissues, which are as well dependent on the mentioned factors. Additionally, applying loads using clinical frequencies, trajectories and magnitudes will result in realistic models which will not overestimate the survival of the samples. Another critical factor to be considered is the lubrication of the sample during biomechanical studies to avoid an excessive and unrealistic wear on the tissues of study.

Therefore, a correct dental biomechanical model should consider at least various of these aspects to resemble the realistic biomechanical behavior and response of the tooth-PDL-bone complex. Experimental dental biomechanical models should avoid studying strictly vertical loads and start applying functional or parafunctional loads with clinically accurate parameters. A critical aspect such as humidity and salivary lubrication needs to be considered due to its influence on dental wear, which can lead to underestimations on the real lifespan of the studied tooth. The behavioral expectations need to be fitted for the studied samples. For that purpose, a correct sample characterization (age, gender, dental condition, dental sample, etc.) should be performed. A significant difference between samples could lead to great variations in the obtained results.

A correct FEA dental biomechanical model involves the definition of the mechanical properties of the dental tissues. These mechanical properties are dependent on the sample of study which should be correctly characterized (mechanically and biologically) and the degree of simplification of the FEA model. While the influence of this simplifications is yet to be studied, a minimal requirement would be to implement the mechanical properties corresponding to the case of study of every tissue present in the tooth-PDL-bone complex. As for the boundary conditions of the model, loads, contacts and load application methods should resemble realistic parameters, biomechanical behaviors and tissue-tissue interactions.

## 16. Future Perspectives and Discussion

While simplifying a biomechanical model does not necessarily mean that the results are not valid and do not represent the natural behavior of the real model, oversimplifications may sum little variations that could end up in results that divert from that natural behavior. Ordinola-Zapata et al. (2022) recently presented a critical analysis on the methodologies of biomechanical studies on root filled teeth, concluding that biomechanical studies in the current literature have results dependent on the model assumptions and this dependence should be reduced in the design phase of the experiment [225].

From the authors’ point of view, this dilemma should be tackled through two different paths. The first path involves avoiding oversimplifications of the model. This can be achieved designing experimental and FEA biomechanical models that mimic the natural behavior and environment of teeth. Real jaw trajectories, which are not completely vertical nor extremely angulated should be reproduced in cases that involve occlusal contact. While the trajectories and load directions of several parafunctional loads (grinding and clenching) have been reported, their influence on teeth with time is not yet completely clear, being necessary to address these events in fatigue-evaluating studies. Both loading frequency and number of cycles are vital parameters that permit an interpretation of the teeth survivability through the years under different conditions, and they should also reflect the natural behavior of the jaw. Physiological and environmental parameters such as age, gender, lubrication, or habits should also be considered to mimic natural human behavior. Taking these factors into account, simplified models can move closer to the natural behavior of teeth.

The second path involves a validation model that serves as contrast for former simplified models. Ideally, this can only be achieved in healthy functioning teeth; however, in vivo fatigue studies are not viable. Hence, future studies should try to reproduce the natural loading conditions of teeth to have a broader vision on the mechanical response of teeth in their natural environment. For this purpose, post mortem studies should be considered to validate the results obtained in in vitro studies due to the better reproducibility of natural conditions, although taking into account the ethical implications involved with this type of samples.

Additionally, fatigue studies should pose a major role when studying new dental biomaterials or treatments, as they are essential for determining the future lifespan of the tooth.

## Figures and Tables

**Table 1 materials-15-07852-t001:** Enamel Mechanical Properties: sample characterization, methodology used and results obtained from the different studies.

Author(s)	Sample Caracterization		Methodology	Results		
	Tooth/Patient	Enamel Selection and Preparation		Elastic Modulus (GPa)	Compressive Strength (MPa)	K_c_ (MPa m^0.5^)
Craig et al. (1961) [14]	Freshly extracted mandibular molars (*n* = 12)	Enamel samples: 1/32 inch diameter	Compression test	77.9 ± 4.8 (Side) 84.1 ± 6.2 (Cusp)	384.5 ± 101.9 370.8 ± 87.6	-
	Enamel in cusps and lateral surfaces showed similar compressive properties.					
Staines et al. (1981) [40]	Human teeth	Wet and dried samples mounted in epoxy resin blocks	Indentation	83	-	-
	Elastic modulus varied with moisture content and enamel orientation.					
Mahoney et al. (2000) [41]	Primary maxillary first molars (age: 4–7 y; *n* = 8)	Teeth mounted in epoxy resin blocks were grinded and polished	Ultra-micro-indentation (UMI)	80.9 ± 6.7 79.8 ± 8.9	-	-
	UMI is a potential alternative method for measuring elastic modulus and hardness.					
Bajaj et al. (2008) [16]	Third Molars (age:17–27 y; *n* = 8)	Sections from cuspal region oriented with prisms parallel to the plane of crack growth	Fatigue crack growth test	-	-	0.9
	Microstructural arrangement of the prisms promotes exceptional resistance to crack growth.					
Park et al. (2008) [2]	Third Molars (age groups: ‘‘young’’ (18 ≤ age ≤ 30; *n* = 7)/‘‘old’’ (55 ≤ age; *n* = 7)	Teeth mounted in polyester resin foundation were sliced and polished	Nanoindentation	75 (Young,Inner) 82 (Young,Middle) 87 (Young,Outer) 79 (Old,Inner) 90 (Old,Middle) 100 (Old,Outer)	-	0.88 0.88 0.92 0.88 0.73 0.67
	Elastic modulus and hardness increased with distance from the DEJ regardless of patient age.					
Ang et al. (2009) [42]	Third Molar (*n* = 1)	Tooth crown cut and glued to steel core	Nanoindentation	123	-	-
	Understanding the elastic–plastic transition is relevant due to the irreversible wear and fatigue that occur past this transition.					
Bajaj and arola (2009) [43]	Third Molars (age: 17–22; *n* = 6)	Small cubes (2 × 2 × 2 mm) of cuspal enamel	Fatigue crack growth test	-	-	2.04 ± 0.23
	The microstructure of enamel in the decussated region promotes crack growth toughness approximately three times higher than dentin and over ten times higher than bone.					
Arola et al. (2010) [39]	-	-	Review	70–110	-	0.7–0.21
	Many challenges for fatigue characterization of hard tissues can be attributed to their size and the complexity of their microstructure.					
Chai et al. (2011) [44]	Molars (*n* = 23)	No preparation (*n* = 7)/ Slice 1 mm (*n* = 16)	Contact loading test	-	-	1.02
	A transition from chipping to splitting occurs at higher loads for contacts nearer the central axis of the tooth.					
Zheng et al. (2013) [45]	Third Molars (age groups: “young” (18–25); *n* = 15; “old” (≥55); *n* = 15)	Sections (height = NP) perpendicular to the buccolingual direction	Microindentation	99.47 ± 1.57 (Old) 93.24 ± 2.00 (Young)	-	1.00 ± 0.14 (Outer) 1.23 ± 0.16 (Middle) 1.22 ± 0.21 (Inner) 1.11 ± 0.12 (Outer) 1.27 ± 0.22 (Middle) 1.23 ± 0.23 (Inner)
	Enamel becomes more prone to cracks with aging partly due to the reduction in the interprismatic organic matrix observed with the maturation of enamel.					
Chai (2014) [46]	Molars (age: 20–30; *n* = 6)	-	Bilayer Test	-	-	0.94 ± 0.24
	Stress–strain curve is highly nonlinear due to plastic shearing of protein between and within enamel rods.					
Elfallah et al. (2015) [47]	Third Molars (age: 18–40; *n* = 24)	Embedded in epoxy resin, polished. Half were treated with bleaching agents	Ultra-micro-indentation	-	-	1.3 ± 0.5 (control) 0.8 ± 0.3 (HP) 0.7 ± 0.2 (CP)
	Tooth bleaching agents can produce detrimental effects on the mechanical properties of enamel, possibly as a consequence of damaging or denaturing protein components.					
Yahyazadehfar et al. (2016) [48]	Third Molars (age groups: “young” (17 ≤ age ≤ 25; *n* = 10) and “old” (age ≥ 55; *n* = 10)	Small cubes (2 × 2 × 2 mm) of cuspal enamel	Fatigue crack growth test	-	-	2.05 ± 0.19 (Young,Longitudinal) 1.38 ± 0.35 (Old,Longitudinal) 1.23 ± 0.20 (Young, Transverse) 0.37 ± 0.15 (Old,Transverse)
	Reduction in fracture resistance is attributed to a decrease in the extrinsic toughening capacity.					
Shimomura et al. (2019) [49]	Third Molars	Cuspal region of the tooth cut horizontally and polished with silicon carbide paper	Nanoindentation	60 (Mapping) 100 (Quasi-static) 130–150 (Loading)	-	-
	The elastic–plastic transition point and elastic modulus value increased with substantially increased quasi-static loading strain rate.					
Niu et al. (2019) [50]	Third Molars (age 25–27/female/*n* = 6)	Sections (3.1 × 2.2 × 1.8 mm) of occlusal enamel	Resonant Ultrasound Spectroscopy (RUS)	71.7 ± 7.34	-	-
	RUS could provide precise measurement of elastic properties of dental materials.					
Dejsuvan et al. (2021) [5]	Deciduous molars (age: 4–12)	Sectioned to obtain small enamel samples	Nanoindentation	76.46 ± 10.46 (Low caries) 61.29 ± 13.33 (High caries)	-	-
	The outer enamel of the low caries experience group had greater mechanical properties than did that in the high caries experience group.					
Handbook of nanoindentation with biological application [51]	-	-	-	83.4 ± 7.1 to 105.2 ± 1.3	384	-
Craig’s restorative materials 13 ed. [52]	-	-	-	84	-	-

**Table 2 materials-15-07852-t002:** Dentin Mechanical Properties: sample characterization, methodology used and results obtained from the different studies.

Author(s)	Sample		Methodology	Results	
	Tooth/Patient	Dentin Selection and Preparation		Elastic Modulus (GPa)	Compressive Strength (MPa)
Peyton et al. (1952) [15]	First and Second Molars (*n* = 10)	Sectioned to obtain small cylindrical (ϕ = 1.8 mm; length = 4.5 mm) dentin samples	Compression test	11.51	248
	The physical properties of dentin are influenced by physiological differences, directional effects in the tooth structure, rate of stress application, ratio length/diameter of the test specimen and soundness of test specimen.				
Craig and Peyton (1958) [4]	First and Second Molars (*n* = 9)	Sectioned to obtain small cylindrical (ϕ = 0.1 inch; length = 0.1–0.3 inch) dentin samples	Compression test	16.5–18.6	275–300
	The total deformation below the proportional limit consisted of pure and retarded elastic deformation.				
Watts et al. (1987) [53]	Lower Molars (*n* = 35)	Sectioned to obtain small (3 × 2.5 × 1.25 mm) dentin samples	Compression test	13.26	260
	A statistically significant, linear regression relationship was found between modulus (E) and temperature. The higher the temperature, the lower E.				
Mahoney et al. (2000) [41]	Decidious maxillary first molars (age: 4–7; *n* = 8)	Teeth were mounted in epoxy resin blocks, grinded and polished	UMI	20.55 ± 2 (50 mN) 19.2 ± 1.84 (150 mN)	-
	UMI is a potential alternative method for measuring elastic modulus and hardness.				
Kinney et al. (2003) [10]	-	-	Review	18–20	-
	The elastic properties of dentin depend on the microstructure of the intertubular dentin matrix.				
Yan et al. (2008) [20]	Third Molars (*n* = 10)	Sectioned to obtain small (1.6 × 1.6 × 10 mm) dentin samples	Three-point flexure test	15 ± 0.5 (In-plane parallel) 15.4 ± 0.4 (Anti-plane parallel)	-
	The J integral of anti-plane parallel specimens is significantly greater than that of in-plane parallel specimens.				
Arola et al. (2010) [39]	-	-	Review	12–20	-
	Many challenges for fatigue characterization of hard tissues can be attributed to their size and the complexity of their microstructure.				
Young June Yoon (2013) [54]	-	-	Speed of Sound and Resonant Ultrasound Spectroscopy	20.7–25.4	-
	The use of these methodologies gives results similar to former studies. These methodologies could be used to further study the properties of dentin.				
Rodrigues et al. (2018) [55]	Irradiated and non-irradiated Third Molars (*n* = 10)	Crowns were sectioned and divided in halves	Indentation	17.18 ± 1.64 (Non-irradiated, Superficial) 17.88 ± 0.92 (Non-irradiated, Middle) 18.34 ± 1.58 (Non-irradiated, Deep) 14.20 ± 0.66 (Irradiated, Superficial) 13.95 ± 1.05 (Irradiated, Middle) 14.36 ± 1.46 (Irradiated, Deep)	-
	Elastic modulus decreased after irradiation of samples.				
Muslov (2018) [56]	-	-	Mathematical formulation	23.3	-
	Dentin and tooth enamel are not isotropic media due to the symmetry of their mineral component hydroxyapatite crystals.				
Craig’s restorative materials 13 ed. [52]	-	-	-	17	297

**Table 3 materials-15-07852-t003:** Results of Dentin Fatigue Properties.

Author(s)	Sample		Methodology		Results		
	Tooth/Patient	Dentin Selection and Preparation		∆K_th_ (MPa m^0.5^)	K_c_ (MPa m^0.5^)	Paris Exponent (m)	Paris Coefficient (C)
Iwamoto and Ruse (2003) [58]	Molars (*n* = 24)	Sectioned to obtain dentin triangular prisms (4 × 4 × 8 mm)	Notch Triangular Prism Test	-	1.97 ± 0.17 (Parallel) 1.13 ± 0.36 (Perpendicular)	-	-
	Both the hypermineralized peritubular dentin and the orientation of collagen fibrils surrounding the tubules could be responsible for the significant differences in K_C_.						
Nalla (2003) [6]	Molars	Sectioned to obtain small (0.9 × 0.9 × 10 mm) dentin samples	Fatigue Cantilever Test	1.06	1.8	8.76	6.24 × 10^−11^
	The presence of small (on the order of 250 µm) incipient flaws in human teeth will not radically affect their useful life						
Bajaj et al.(2006) [17]	Second and Third molars (“young” (18–35); *n* = 8; “old” (≥47); *n* = 14)	Sectioned to obtain small (6 × 4 × 2 mm) dentin samples	Fatigue crack growth test	0.7 (Young) 0.5 (Old)	-	13.3 ± 1.1 21.6 ± 5.2	1.76 × 10^−5^ 2.90 × 10^−2^
	The fatigue crack growth resistance of human dentin decreases with age and dehydration.						
Zhang et al. (2007) [18]	Second and Third molars	Sectioned to obtain small (2 × 2 × 2 mm) dentin samples	Fatigue crack growth test	-	1.65 (Young) 1.3 (Old)	13.8 ± 7.6	-
	Future evaluations of fracture and the mechanisms of toughening in these materials should account for the contributions of inelastic deformations.						
Yan et al. (2008) [20]	Third Molars (*n* = 10)	Sectioned to obtain small (1.6 × 1.6 × 10 mm) dentin samples	3-point Flexure Test	-	2.4 ± 0.2 (In-plane parallel) 2.5 ± 0.2 (Anti-plane parallel)	-	-
	The J integral of anti-plane parallel specimens is significantly greater than that of in-plane parallel specimens.						
Nazari et al. (2009) [59]	Third Molars (“young”) (18–35); “middle” (35–55; “old” (≥55)) (*n* = 14)	Sectioned to obtain small (6 × 4 × 2 mm) dentin samples	Fatigue crack growth test	-	1.65 ± 0.1 (Young) 1.43 ± 0.1 (Middle) 1.17 ± 0.09 (Old)	-	-
	Human dentin exhibits a rising R-curve. There is a significant reduction in both the initiation and plateau components of toughness with age.						
Yan et al. (2009) [21]	Third Molars (*n* = 16)	Sectioned to obtain small (1.6 × 1.6 × 10 mm) dentin samples	3-point Flexure Test	-	2.2 ± 0.2 (In-plane parallel) 2.4 ± 0.2 (Anti-plane parallel)	-	-
	Human dentin has a fracture surface similar to those of brittle materials.						
Arola et al. (2010) [39]	-	-	Review	0.6	1.5–2.1	10–20	-
	Many challenges for fatigue characterization of hard tissues can be attributed to their size and the complexity of their microstructure.						
Ivancik et al. (2011) [60]	Third Molar (*n* = 31)	Sectioned to obtain small (6 × 4 × 2 mm) dentin samples	Fatigue crack growth test	0.8 ± 0.12 (Deep) 1.0 ± 0.06 (Middle) 1.2 ± 0.08 (Peripheral)	-	27.5 ± 7 25.5 ± 2.9 26.7 ± 2.8	1.64 × 10^−5^ ± 0.02 × 10^−5^ 4.41 × 10^−8^ ± 0.21 × 10^−8^ 5.61 10^−10^ ± 0.12 × 10^−10^
	Molars with deep restorations are more likely to suffer from cracked-tooth syndrome, because of the lower fatigue crack growth resistance of deep dentin						
Ivancik et al. (2012) [57]	Third Molars (“young” (17–33); *n* = 32; “old” (≥55); *n* = 15)	Sectioned to obtain small (6 × 4 × 2 mm) dentin samples	Fatigue crack growth test	1.03 ± 0.1 (Young, Parallel) 0.83 ± 0.1 (Young,Perpendicular) 0.77 ± 0.1 (Old,Parallel) 0.60 ± 0.1 (Old,Perpendicular)	-	25.47 ± 3.0 14.15 ± 1 23.11 ± 5.1 24.16 ± 4.3	4.79 × 10^−8^ 2.69 × 10^−5^ 6.61 × 10^−5^ 1.58 × 10^−2^
	Regardless of tubule orientation, aging of dentin is accompanied by a significant reduction in the resistance to the initiation of fatigue crack growth, as well as a significant increase in the rate of incremental extension.						
Orrego et al. (2017) [19]	Third Molars (age = 17–33; *n* = 103)	Sectioned to obtain small (1.5 × 0.5 × 10 mm) dentin samples	4-point Cyclic Flexure Test	1.03 ± 0.06 (Middle) 1.23 ± 0.08 (Outer)	-	25.5 ± 2.9 26.7 ± 2.8	4.4 × 10^−8^ 5.6 × 10^−10^
	The endurance limit after biofilm exposure was 60% lower than that of the control environment.						

**Table 4 materials-15-07852-t004:** Data for PDL Mechanical Properties.

Author(s)	Sample		Methodology	Results	
	Tooth/Patient	PDL Selection and Preparation		Elastic Modulus (MPa)	Poisson Ratio
Thresher and Saito (1973) [64]	Upper Incisor	-	2D FEM	1379	0.45
	FEM is an appropriate analysis tool for the study of teeth.				
Mandel et al. (1986) [8]	Lower First Premolars (age:23–55/male/*n* = 20)	Teeth were cut is small samples (height = 1.05 mm) containing dentin, PDL and alveolar bone	Flexural deformation	3	-
	In order to compare the mechanical properties of PDL care should be taken to compare areas at the same root level.				
Rees and Jacobsen (1997) [65]	Lower First Premolar (*n* = 1)	Tooth was embedded in epoxy resin and sectioned bucco-lingually through the centre of the tooth	2D FEM + Uniaxial tensile test	50	0.49
	An elastic modulus of 50 MPa gave good correlation between the finite element model and the experimental systems.				
Jones et al. (2001) [66]	Upper Incisors (age = 24.7–36.5; *n* = 10)	-	3D FEM + In vivo compressive load test	1	0.45
	PDL is the main mediator of orthodontic tooth movement.				
Yoshida et al. (2001) [67]	Upper Incisor (age = 24–27/female/*n* = 2)	-	In vivo compressive load test	0.12–0.96	-
	The values of Young’s moduli increased almost exponentially with the increment of load due to a non-linear elasticity of the PDL.				
Wu et al. (2018) [7]	Lower Incisors (age = 31–52/male/*n* = 3)	Teeth were cut is small samples (8 × 6 × 2 mm) containing dentin, PDL and alveolar bone	Uniaxial tensile test	0.33–6.82	-
	The elastic behavior of the PDL is infuenced by the loading rate, tooth type, root level, and individual variation.				

**Table 5 materials-15-07852-t005:** Data for Mandible and Maxilla Mechanical Properties.

Author(s)	Site		Methodology	Results	
	Tooth/Patient	Bone Selection and Preparation		Elastic Modulus (GPa)	Poisson Ratio
Cortical					
Borchers and Reichart (1983) [84]	-	-	FEM	13.7	0.3
	Presence in the model of a connective tissue layer around the implant reduces stress peaks.				
Schwartz-Dabney and Dechow (2002) [80]	Edentolous Mandibles (age = 58–88; *n* = 10)	Small cylindrical samples (ϕ = 4 mm) were sectioned from the mandibles	Ultrasonic Waves Emission	E_1_ = 13.26 E_2_ = 17.51 E_3_ = 26.28	µ_12_ = 0.25 µ_31_ = 0.45 µ_23_ = 0.22
	Mandibular cortical bone in edentulous mandibles differs from that of dentate mandibles in cortical thickness, elastic and shear moduli, anisotropy, and orientation of the axis of maximum stiffness.				
Lettry et al. (2003) [85]	Mandibles (Premolar and Molar Sections) (age = 53–106; *n* = 5)	Small prismatic (section = 1 × 2 mm) samples were sectioned from the mandibles	3-point Bending Test	5–15 (Approximation)	-
	A weak correlation was found between the elastic modulus values and the computer tomography number of the mandible.				
Seong et al. (2009) [86]	Fresh Edentulous Maxilla and Mandibles (*n* = 4; age = 72–91)	Samples were sectioned (width = 3 mm) from the maxillas and mandibles in different areas	Nanoindentation	16.8 (Anterior) 19.7 (Posterior)	-
	Bone physical properties differ between regions of the maxilla and mandible. Generally, mandible has higher physical property measurements than maxilla				
**Trabecular**					
Borchers and Reichart (1983) [84]	N.A.	N.A.	FEM	1.37	0.3
	Presence in the model of a connective tissue layer around the implant reduces stress peaks.				
Misch et al. (1999) [87]	Fresh Mandibles (*n* = 9; age = 56–90)	Small cylindrical samples (ϕ = 5 mm; length = 5 mm) were sectioned from the mandibles	Compression Test	67.48 (Anterior) 47.3 (Premolar) 35.55 (Molar)	-
	Trabecular bone in the mandible possesses significantly higher density, elastic modulus, and ultimate compressive strength in the anterior region than in either the middle or distal regions.				
O’mahony et al. (2000) [79]	Fresh Edentulous Mandible (*n* = 1/female/age = 74)	Samples (4.4 × 4.4 × 4.8 mm)were sectioned from the mandible	Compression Test	0.91 (Mesio-Distal) 0.51 (Bucco-Lingual) 0.11 (Infero-Superior)	-
	Models of cancellous bone in the jaw should present a symmetry axis along the infero-superior (weakest) direction.				
Seong et al. (2009) [86]	Fresh Edentulous Maxilla and Mandibles (*n* = 4; age = 72–91)	Samples were sectioned (width = 3 mm) from the maxillas and mandibles in different areas	Nanoindentation	16.8 19.7	-
	Bone physical properties differ between regions of the maxilla and mandible. Generally, mandible has higher physical property measurements than maxilla				

**Table 6 materials-15-07852-t006:** Summary of experimental and FEM studies on biting loads.

Author(s)	Field of Study and Fatigue/Fracture Criteria	Load/Cycles	Geometry	Type of Study
Sakaguchi et al. (1992) [101]	Fracture Mechanics/Forman crack growth rate	13.9 N	Maxillary premolar	2D FEM
Libman and Nicholls (1995) [134]	Restorations and Crowns/Fracture Appearance	40 N/Until fracture at 1.2 Hz	Central incisors	Experimental
Cohen et al. (1997) [135]	Endodontic Posts/-	22.2 N/4 × 10^6^ cycles at 3 Hz	Premolars and Incisors	Experimental
Lanza et al. (2005) [136]	Endodontic Posts/Material Failure Limits	10 N at 125°	Scanned maxillary incisor	3D FEM
Dejak et al. (2006) [107]	Fracture Mechanics/Tsai-Wu Ratio	0 to 200 N	2-D Finite element model of mandibular molar and crown of maxillary antagonizing molar	2D FEM
Cobankara et al. (2008) [137]	Restorations and Crowns/Fracture Appearance	50 N/6 × 10^4^ cycles at 1.3 Hz24 h later, 1 mm/min compressive load	Mandibular molars	Experimental
Lin et al. (2010) [138]	Restorations and Crowns/Failure Probability through Weibull Analysis	50 N/2 × 10^4^ cycles at 3 Hz (Experimental) 100 N (FEM)	Maxillary premolars	Experimental + 3D FEM
Uy et al. (2010) [139]	Restorations and Crowns, Endodontic Posts/Strain Amplitude	58.8 N at 135°	First and second premolars	Experimental
Barani et al. (2011) [140]	Fracture Mechanics/Fracture Appearance (Experimental) and Critical Energy Release Rate (XFEM)	Compression test with indenter	Molars (Experimental), 3-D Dome structure (XFEM)	Experimental + 3D FEM
Du et al. (2011) [141]	Endodontic Posts/-	100 N at 45°	Lower first premolar	3D FEM
Rodríguez-Cervantes et al. (2011) [142]	Endodontic Posts/-	0–50 N/1.2 × 10^6^ cycles	Scanned premolars and modelled PDL and bone	3D FEM
Nie et al. (2012) [104]	Restorations and Crowns/Fracture Appearance	127.4 N at 45°/1.2 × 10^6^ cycles at 6 Hz	Lower premolars	Experimental
Benazzi et al. (2014) [143]	Stress Distributions/-	100 N	Lower second premolar	3D FEM
Toledano et al. (2014) [144]	Restorations and Crowns/-	49 N/1 × 10^5^ cycles	Halves of third molars	Experimental
Toledano et al. (2015) [105]	Adhesives/-	225 N/259,200 cycles at 3 Hz	Third molars	Experimental
Vukicevic et al. (2015) [22]	Restorations and Crowns/Fatigue Failure Index, Stress Ratio, Goodman’s Line, Paris Law	100, 150 and 200 N/1 × 10^6^ cycles	Maxillary second premolars	3D FEM
Zhu et al. (2015) [145]	Restorations and Crowns/Fracture Appearance or Fluid Penetration	260 N/2 × 10^6^ cycles at 4 Hz	Upper first premolars	Experimental
Gao et al. (2016) [146]	Fracture Mechanics/Maximum Strain Energy (Zhang (2011) criteria)	100–700 N/1 × 10^6^ cycles at 2 Hz	Third molars	Experimental
Toledano et al. (2016) [147]	Adhesives/Fracture Appearance	225 N/259,200 cycles at 3 Hz	Third molars	Experimental
Ossareh et al. (2018) [23]	Fracture Mechanics/Fracture Appearance	50 N/1.2 × 10^6^ cycles at 1.6 Hz w/6 × 10^3^ 2 min cycles × 5 °C/55 °C (Experimental) 100 N (FEM)	Mandibular premolars	Experimental + 3D FEM
Chen et al. (2021) [148]	Restorations and Crowns/Fracture Appearance	50 N/1.2 × 10^6^ cycles w/2 × 10^4^ cycles × 5 °C/55 °C (Experimental)	Maxillary premolars	Experimental + 3D FEM
Chen et al. (2021) [149]	Restorations and Crowns/Fracture Appearance	50 N at 45° (tongue direction)/1.2 × 10^6^ cycles at 2 Hz. (Experimental)50 N oblique compressive load (FEM)	Maxillary premolars	Experimental + 3D FEM

**Table 7 materials-15-07852-t007:** Summary of experimental and FEM studies on clenching loads.

Author(s)	Field of Study and Fracture/Fatigue Criteria	Load/Cycles	Geometry	Type of Study
Kovarik et al. (1992) [161]	Endodontic Posts/Fracture appearance	340 N/1× 10^6^ cycles at 1 Hz	Canines	Experimental
Rees (2002) [162]	Stress distributions/Maximum Principal Stress	500 N	Lower second premolar	2D FEM
Maceri et al. (2007) [163]	Endodontic Posts/Rankine Stress	400 N (Vertical) 200 N (45°)	Lower premolar	3D FEM
Hayashi et al. (2008) [164]	Endodontic Posts/Fracture appearance	90° at 0.5 mm/min (Static, Upper premolar) 45° at 0.5 mm/min (Static, Lower premolar) 90° (Fatigue, Upper premolar) 45° (Fatigue, Lower premolar) 2 × 10^6^ cycles at 2 Hz	Upper and lower premolars	Experimental
Cheng et al. (2009) [165]	Stress distributions/-	500 N (0°, 30°, 45° and 60°)	Simulated canals	3D FEM
Magne et al. (2010a) [166]	Restorations and Crowns/Fracture Appearance	200 N/5 × 10^6^ 3 cycles at 5 Hz Then 200 N increasing steps until 1200 N/Max. 3 × 10^4^ cycles/step at 5 Hz	Maxillary molars	Experimental
Magne (2010b) [167]	Restorations and Crowns/Maximum Principal Stress	200 N, 700 N	Mandibular molar	3D FEM
Inoue et al. (2011) [168]	Material Sciences/Fracture Appearance	5 MPa steps/1 × 10^5^ cycles (Staircase method)	Bovine lower central incisors	Experimental
Kasai et al. (2012) [169]	Stress Distributions/-	100 N, 200 N and 800 N	Mandible w/two implants in the molar region Mandible w/four implants in the pre- and molar regions	3D FEM
Nie et al. (2012) [104]	Restorations and Crowns, Endodontic Posts/Fracture Appearance	Increasing load at 45° until fracture is detected	Lower premolars	Experimental
Magne et al. (2014) [24]	Restorations and Crowns/Fracture Appearance	200 N/5 × 10^3^ cycles Then 200 N increasing steps until 1400 N/Max. 3 × 10^4^ cyles/step at 10 Hz	Maxillary molars	Experimental
Jayasudha et al. (2015) [102]	Stress distributions/Maximum Principal Stress	Sinusoidal 800 N/1 cycle for 4 ms	Incisor	3D FEA
Kayumi et al. (2015) [170]	Stress Distributions/-	40 N, 100 N, 200 N, 400 N and 800 N	Mandible w/ eight implants in the pre- and molar regions	3D FEM
Toledano et al. (2015) [105]	Adhesives/-	225 N/6171 cycles at 0.072 Hz (Cyclic Clenching) 225 N/For 24 h and 72 h (Permanent Clenching)	Third molars	Experimental
Gao et al. (2016) [146]	Fracture Mechanics/Maximum Strain Energy (Zhang (2011) criteria)	100 N-700 N/Until reaching critical displacement at 2 Hz	Third molars	Experimental
Toledano et al. (2016) [147]	Adhesives/Fracture Appearance	225 N/6171 cycles at 0.072 Hz (Cyclic Clenching) 225 N/For 24 h and 72 h (Permanent Clenching)	Third molars	Experimental
Magne and Cheung (2017) [103]	Stress Distributions/-	500 N	Maxillary first molar	3D FEM
Missau et al. (2017) [171]	Fracture Mechanics/Fracture Appearance	200 N/5 × 10^3^ cycles Then 100 N increasing steps until 900 N/Max. 3 × 10^4^ cyles per step at 10 Hz	Canines	Experimental
Da Fonseca et al. (2018) [172]	Restoration and Crowns/-	300 N (Occlusal) 300 N (Oblique)	Maxillary premolar	3D FEM
Yoon et al. (2018) [173]	Endodontic Posts/-	300 N	Mandibular first molar	3D FEM
Dartora et al. (2019) [25]	Restoration and Crowns/Fracture Appearance and Mohr-Coulomb stress	200 N/5 × 10^3^ cycles Then 200 N increasing steps until 2800 N/Max. 1 × 10^4^ cycles/step at 20 Hz (Experimental) 300 N at 30° (tongue long-axis) (FEM)	Mandibular molars	3D FEM and Experimental
Wan et al. (2019) [174]	Fracture Mechanics/Maximum Principal Stress	0,2 mm displacement	Human premolars	3D FEM
Fráter et al. (2021) [175]	Endodontic Posts/Fracture Appearance	100–500 N/2.5 × 10^4^ cycles at 5 Hz 600–1000 N/3 × 10^5^ cycles at 5 Hz	Upper premolars	Experimental
He et al. (2021) [176]	Restoration and Crowns/-	600 N (Occlusal) + 20 N (bucco-lingual)	First mandibular molar	3D FEM
Kim et al. (2021) [106]	Restoration and Crowns/Maximum Principal Stress and Von Mises	1000 N	Lower first molar	3D FEM
Meng et al. (2021) [177]	Restoration and Crowns/-	600 N	Mandibular molars	3D FEM
Zheng et al. (2021) [178]	Restoration and Crowns/-	200 N Calculations from 300 N to 1500 N in proportion to 200 N results	Mandibular molar	3D FEM and Statistical
Lin et al. (2022) [26]	Fracture Mechanics/Fracture Appearance, Fracture Probability and Cumulative Damage and S-N curve	100 N Increasing steps until 400 N/3 × 10^3^ cycles/step Then 50 N increasing steps until 850 N/1 × 10^3^ cycles/step (Experimental) Static loads at each step (FEM)	Mandibular premolars	3D FEM and Experimental

**Table 8 materials-15-07852-t008:** Summary of experimental and FEM studies on grinding loads.

Author(s)	Field of Study	Loads	Geometry	Type of Study
Kaidonis et al. (1998) [194]	Materials Science	32 N, 99.5 N, 162 N/1.33 Hz	Premolars	Experimental
Dejak et al. (2006) [107]	Fracture Mechanics	0–200 N with 0.1 mm medial and lateral displacement	2D Mandibular molar and crown of maxillary antagonizing molar	2D FEM
Yang et al. (2016) [28]	Stress Distributions	908 N, 1470 N, 1960 N, and 2205 N in recorded incision direction	Jaw model	3D FEM
Magne and cheung (2017) [103]	Stress Distributions	500 N	Maxillary first molar	3D FEM
Ortún-terrazas et al. (2020) [29]	Materials Science	Recorded reaction forces on the lower left cuspid tooth of the full dentition model	Jaw model and Incisive	Experimental + 3D FEM
Sagl et al. (2022) [27]	Biomechanics	3 mm lateral excursion with 6 different inclinations	Jaw model	3D FEM

## Data Availability

Not applicable.

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
