# Peer review of "Biomechanical Modelling for Tooth Survival Studies: Mechanical Properties, Loads and Boundary Conditions—A Narrative Review"

_materials, 2022, doi:10.3390/ma15217852_

Round 1

Reviewer 1 Report

The authors reported that their review aimed "to identify the most relevant factors related to tooth survival that should be implemented in future dental biomechanical modeling studies in terms of mechanical properties of the complex tooth/PDL//bone, functional and parafunctional loads, and boundary conditions." 

The results obtained by previous studies regarding enamel, dentin, PDL, Mandible, and Maxilla mechanical properties have been summarized in 5 different tables. 

The authors have also analyzed and discussed the different methodologies used to perform fatigue, or bending tests on structural tissues (enamel, dentin PDL, and bone). Additionally, other conditions, including functional (biting) and parafunctional (clenching and grinding) loads and different physical and physiological factors, have been taken into account to understand the biomechanical behavior of the teeth.

However, the conclusions have not been clearly described and are not completely coherent with the aim of this review.

Author Response

Reviewer #1

The authors reported that their review aimed "to identify the most relevant factors related to tooth survival that should be implemented in future dental biomechanical modeling studies in terms of mechanical properties of the complex tooth/PDL//bone, functional and parafunctional loads, and boundary conditions."

The results obtained by previous studies regarding enamel, dentin, PDL, Mandible, and Maxilla mechanical properties have been summarized in 5 different tables.

The authors have also analyzed and discussed the different methodologies used to perform fatigue, or bending tests on structural tissues (enamel, dentin PDL, and bone). Additionally, other conditions, including functional (biting) and parafunctional (clenching and grinding) loads and different physical and physiological factors, have been taken into account to understand the biomechanical behavior of the teeth.

Authors: Thank you very much for your kind words.

However, the conclusions have not been clearly described and are not completely coherent with the aim of this review.

Authors: Thank you for pointing this out. We have changed the conclusions to be coherent with the aim of our study.

Reviewer 2 Report

The review is well prepared from a physical point of view, however the biological point of view is missing. Therefore, clinical conclusions are rather invalid. Please add the correct sample characterization: Deciduous vs. permanent enamel, dentin, cementum, PDL, yaw bones etc. Young vs. old enamel, abraded, matured, prismatic, prismless. Dentin healthy, young vs.old; dentin sclerotic, pulpless etc. etc. Please organize the Tables 1 - 5 in comparable sections like Sample (correct description), Methodology, Results, Consequences.

In Conclusions please describe your model proposals, what is indeed clinically relevant? Using human or animal teeth, the ethical aspects shoulb be mentioned

Author Response

The review is well prepared from a physical point of view; however the biological point of view is missing. Therefore, clinical conclusions are rather invalid. Please add the correct sample characterization: Deciduous vs. permanent enamel, dentin, cementum, PDL, yaw bones etc. Young vs. old enamel, abraded, matured, prismatic, prismless. Dentin healthy, young vs.old; dentin sclerotic, pulpless etc. etc. Please organize the Tables 1 - 5 in comparable sections like Sample (correct description), Methodology, Results, Consequences.

 Authors: The reviewer would like to thank the reviewer for this comment and for adding the suggestion for a better table organization. All tables have been re- organized following the guidelines form the reviewer and they all read much better now and contain more relevant information that was not included before.  A correct sample characterization, according to information provided in the analyzed studies, has been included in two sub-columns under the column “Sample”, one column corresponding to the studied teeth and patients (Sub-column Teeth/Patient) and other corresponding to the obtention and preparation of the samples prior to the mechanical testing (Sub-column Tissue obtention and preparation). The Consequences column proposed by the reviewer has also been included in Tables 1-5 as a sub-row.

In Conclusions please describe your model proposals, what is indeed clinically relevant?

Authors: As suggested by the reviewer, the conditions proposed for a correct experimental and FEA model have been included in Conclusions section.

Using human or animal teeth, the ethical aspects should be mentioned.

Authors: The ethical aspects have now been mentioned in Future perspectives and discussion. Thank you.

Reviewer 3 Report

Dear Authors,

the paper is really well written, although some of the information (like on dentin) are basic ones, and I would only add some suggestions how to improve its quality by adding some missing aspects, mainly in the discussion section:

1. I would add dome works of one of the best experts' teams in TMD disorders, including bruxism - see the following papers:

Wieckiewicz M, Smardz J, Martynowicz H, Wojakowska A, Mazur G, Winocur E. Distribution of temporomandibular disorders among sleep bruxers and non-bruxers-A polysomnographic study. J Oral Rehabil. 2020 Jul;47(7):820-826. doi: 10.1111/joor.12955.

Martynowicz H, Dymczyk P, Dominiak M, Kazubowska K, Skomro R, Poreba R, Gac P, Wojakowska A, Mazur G, Wieckiewicz M. Evaluation of Intensity of Sleep Bruxism in Arterial Hypertension. J Clin Med. 2018 Oct 5;7(10):327. doi: 10.3390/jcm7100327. PMID: 30301160;

2. I would also add the aspect of influence of COVID pandemic on the problems of bruxism, as it is an iportant aspect in this case:

- Emodi-Perlman A, Eli I. One year into the COVID-19 pandemic – temporomandibular disorders and bruxism: What we have learned and what we can do to improve our manner of treatment. Dent Med Probl. 2021;58(2):215–218. doi:10.17219/dmp/132896

3. I think it would be also valid to add the information on changes seen in teeth and materials in fractal dimension analysis - this is a novel technic used in dentistry, not a typical one:

Skośkiewicz-Malinowska K, Mysior M, Rusak A, Kuropka P, Kozakiewicz M, Jurczyszyn K. Application of Texture and Fractal Dimension Analysis to Evaluate Subgingival Cement Surfaces in Terms of Biocompatibility. Materials (Basel). 2021 Oct 7;14(19):5857. doi: 10.3390/ma14195857.

Grzebieluch W, Kowalewski P, Grygier D, Rutkowska-Gorczyca M, Kozakiewicz M, Jurczyszyn K. Printable and Machinable Dental Restorative Composites for CAD/CAM Application-Comparison of Mechanical Properties, Fractographic, Texture and Fractal Dimension Analysis. Materials (Basel). 2021 Aug 29;14(17):4919. doi: 10.3390/ma14174919.

Those aspects are especially important for the perspectives

In the title it should also be mentioned that was a narrative review.

After that changes, the paper could be accepted

Author Response

Dear Authors,

the paper is really well written, although some of the information (like on dentin) are basic ones, and I would only add some suggestions how to improve its quality by adding some missing aspects, mainly in the discussion section:

  1. I would add dome works of one of the best experts' teams in TMD disorders, including bruxism - see the following papers:

- Wieckiewicz M, Smardz J, Martynowicz H, Wojakowska A, Mazur G, Winocur E. Distribution of temporomandibular disorders among sleep bruxers and non-bruxers-A polysomnographic study. J Oral Rehabil. 2020 Jul;47(7):820-826. doi: 10.1111/joor.12955.

- Martynowicz H, Dymczyk P, Dominiak M, Kazubowska K, Skomro R, Poreba R, Gac P, Wojakowska A, Mazur G, Wieckiewicz M. Evaluation of Intensity of Sleep Bruxism in Arterial Hypertension. J Clin Med. 2018 Oct 5;7(10):327. doi: 10.3390/jcm7100327. PMID: 30301160;

Authors: Thank you very much for your kind words. As suggested by the reviewer, those studies have been referenced now. Please see references nº  185 and 186.

  1. I would also add the aspect of influence of COVID pandemic on the problems of bruxism, as it is an important aspect in this case:

- Emodi-Perlman A, Eli I. One year into the COVID-19 pandemic – temporomandibular disorders and bruxism: What we have learned and what we can do to improve our manner of treatment. Dent Med Probl. 2021;58(2):215–218. doi:10.17219/dmp/132896

Authors: Thank you very much for the recommendation. This study has now been included and referenced (220) in the section “Patient dependent factors”.

  1. I think it would be also valid to add the information on changes seen in teeth and materials in fractal dimension analysis - this is a novel technic used in dentistry, not a typical one:

- Skośkiewicz-Malinowska K, Mysior M, Rusak A, Kuropka P, Kozakiewicz M, Jurczyszyn K. Application of Texture and Fractal Dimension Analysis to Evaluate Subgingival Cement Surfaces in Terms of Biocompatibility. Materials (Basel). 2021 Oct 7;14(19):5857. doi: 10.3390/ma14195857.

- Grzebieluch W, Kowalewski P, Grygier D, Rutkowska-Gorczyca M, Kozakiewicz M, Jurczyszyn K. Printable and Machinable Dental Restorative Composites for CAD/CAM Application-Comparison of Mechanical Properties, Fractographic, Texture and Fractal Dimension Analysis. Materials (Basel). 2021 Aug 29;14(17):4919. doi: 10.3390/ma14174919.

Authors: Thank you for pointing out such interesting studies. While fractal dimension analysis is a novel technique for measuring mechanical properties of materials and surface topography, we have not been able to find enough information of this technique being used to determine the material properties of the biological tissues evaluated in this study (dentin, enamel, PDL and mandible/maxilla). The suggested studies address surface topography and material properties of dental materials, but the present review is only focusing in natural tooth components.

In the title it should also be mentioned that was a narrative review. the reviewers suggestion. After that changes, the paper could be accepted.

Authors: Thank you for this suggestion. The term “narrative review” has been added to the title.

Round 2

Reviewer 2 Report

Thank you for following the referee`s proposals, you improved the MS considerably.